# A Network Analysis of Multiple Myeloma Related Gene Signatures

**DOI:** 10.3390/cancers11101452

**Published:** 2019-09-27

**Authors:** Yu Liu, Haocheng Yu, Seungyeul Yoo, Eunjee Lee, Alessandro Laganà, Samir Parekh, Eric E. Schadt, Li Wang, Jun Zhu

**Affiliations:** 1Sema4, a Mount Sinai Venture, 333 Ludlow St., Stamford, CT 06902, USA; 2Department of Genetics and Genomic Sciences, Icahn School of Medicine at Mount Sinai, 1 Gustave L. Levy Pl, New York, NY 10029, USA; 3The Tisch Cancer Institute, Icahn School of Medicine at Mount Sinai, 1 Gustave L. Levy Pl, New York, NY 10029, USA

**Keywords:** multiple myeloma, Bayesian network, gene signature, prognostic, treatment response

## Abstract

Multiple myeloma (MM) is the second most prevalent hematological cancer. MM is a complex and heterogeneous disease, and thus, it is essential to leverage omics data from large MM cohorts to understand the molecular mechanisms underlying MM tumorigenesis, progression, and drug responses, which may aid in the development of better treatments. In this study, we analyzed gene expression, copy number variation, and clinical data from the Multiple Myeloma Research Consortium (MMRC) dataset and constructed a multiple myeloma molecular causal network (M3CN). The M3CN was used to unify eight prognostic gene signatures in the literature that shared very few genes between them, resulting in a prognostic subnetwork of the M3CN, consisting of 178 genes that were enriched for genes involved in cell cycle (fold enrichment = 8.4, *p* value = 6.1 × 10^−26^). The M3CN was further used to characterize immunomodulators and proteasome inhibitors for MM, demonstrating the pleiotropic effects of these drugs, with drug-response signature genes enriched across multiple M3CN subnetworks. Network analyses indicated potential links between these drug-response subnetworks and the prognostic subnetwork. To elucidate the structure of these important MM subnetworks, we identified putative key regulators predicted to modulate the state of these subnetworks. Finally, to assess the predictive power of our network-based models, we stratified MM patients in an independent cohort, the MMRF-CoMMpass study, based on the prognostic subnetwork, and compared the performance of this subnetwork against other signatures in the literature. We show that the M3CN-derived prognostic subnetwork achieved the best separation between different risk groups in terms of log-rank test *p*-values and hazard ratios. In summary, this work demonstrates the power of a probabilistic causal network approach to understanding molecular mechanisms underlying the different MM signatures.

## 1. Introduction

Multiple myeloma (MM) is the second most common hematological malignancy that derives from the neoplastic transformation and proliferation of plasma cell in bone marrow [1]. Genetic aberrations, particularly genomics translocations involving immunoglobulins, are common [2,3] and are associated with prognosis [4]. These include t(4;14) (the translocation of IgH enhancer and the region of *FGFR3/MMSET* [5]), and the deletion of 17p or 17p13 [6,7]. Other putative prognostic aberrations in literature, such as t(14;16) [8] and 1q21 amplification [9,10], are mixed. More recently, patients with bi-allelic *TP53* inactivation, the amplification (>three copies) of *CKS1B*, and IgL-MYC translocation, have been identified as high-risk for relapse in large scale cohorts [11,12]. However, these findings need further validation in independent studies. 

Until 20 years ago, the treatment of MM was limited to steroids and alkylating agents. However, since the early 2000s, immunomodulatory agents and proteasome inhibitors have been introduced to treat MM. Other new agents against MM over the last decade, such as filanesib, dinaciclib, venetoclax, and daratumuab, have been developed and approved as well, or are still working their way through clinical trials [13,14,15]. As a result, the median survival has improved from 3 years to around 7 years [16,17,18]. Despite these improvements in the treatment of MM patients, roughly 20% of MM patients relapse and become drug resistant in 2 years after initial treatments [19,20]. Thus, while the therapeutic options available are increasing, it remains critical to identify high risk patients early and develop personalized treatment options for them to improve outcomes. 

More than 20 different gene expression-based prognostic/response signatures have been reported for MM in last decade [21,22,23,24,25,26,27,28,29,30,31,32,33,34,35,36,37,38,39,40,41,42]. For example, Shaughnessy et al., applied a log-rank test coupled with stepwise multiple linear discriminant analysis to first derive a 70-gene signature (GEP70), and then selected a subset of these genes to predict prognosis of MM patients [38]. Later, it was found that five genes carried most of the discriminatory power of the 70-gene risk model [43]. Others, such as Kuiper et al., first employed univariate cox regression to select 1093 genes, then combined supervised principal component analysis with simulated annealing to reduce to a 92-gene signature (EMC-92) that was predictive of MM prognosis [31]. Only two genes overlap between GEP70 and EMC-92. This sparse overlap may be attributable to different aspects of myeloma biology. MM is a highly heterogeneous disease with diverse distinguished driver events, including simple mutations, genomic rearrangement, and other structure variants [4,14,44]. Multiple subgroups were proposed based on gene expression and/or genomic profiling [44,45]. Different MM subgroups might not share prognostic gene expression signatures. Recent efforts have been made to combine different gene expression signatures or combine gene expression signatures with clinical data to improve the prognostic stratification of MM patients using simulations [46,47,48]. 

In this study, we analyzed a large MM cohort (the Multiple Myeloma Research Consortium, MMRC study) [49], which contains both mRNA and copy number variation (CNV) profiles. We first conducted extensive QC to identify and correct sample-labeling errors (refer to the workflow in Figure 1). To distinguish potential associations and causal relationships among molecular features, we then constructed a multiple myeloma molecular causal network (M3CN) by integrating gene expression and CNV data using our previously described Bayesian network construction procedure [50]. Next, we collected prognostic and treatment response gene signatures from the literature, and then projected these signatures onto the M3CN to derive a robust prognostic MM subnetwork. Our network analysis of this subnetwork and of M3CN more generally, revealed several key regulators of the previously published prognostic and treatment response signatures. Even though these signatures had minimal overlap, most of them were significantly enriched in subnetworks that contained significant overlaps of gene signatures for prognosis and response. Pathway analysis showed that genes in the cell cycle and metabolic processes were enriched in these subnetworks. To assess the predictive power of the prognostic MM subnetwork we derived from M3CN, we stratified MM patients in an independent cohort, the MMRF-CoMMpass study, based on this subnetwork, and compared the accuracy of this stratification against the prognostic signatures from the literature. The M3CN-derived prognostic subnetwork was shown to achieve the best separation between different risk groups in terms of log-rank test *p*-values and hazard ratios. In summary, our results suggest that the network approach can reveal biological mechanisms unifying the diverse MM signatures.

## 2. Results

### 2.1. QC of the MMRC Dataset for Integrative Network Analysis

Gene expression profiles (*n* = 304, GSE26760), CNV profiles (*n* = 254, GSE26849), and associated clinical data for the MMRC study were downloaded from the GEO database [49]. Sample-labeling errors, including sample mislabeling, swapping, duplication, or contamination frequently occur in such multi-omics datasets [51,52]. Thus, it is critical to perform extensive QC to identify and correct such errors before integrating gene expression and CNV profiles for further analysis. 

In MM, genomic alterations are common [49], and gene expression variations are strongly associated with such alterations [53]. In the MMRC dataset, the expression levels of 8182 genes were significantly associated with CNVs that contained the respective genes in cis form (cis-regulation), with a Benjamini–Hochberg multiple testing corrected *p*-value < 0.01. Probabilistic multi-omics data matcher (proMODMatcher), a computational approach to identify and correct sample-labeling errors in multiple types of omics data [51,52], was applied to match mRNA and CNV profiles in the MMRC datasets. Among 246 pairs of gene expression and the CNV profiles of the common patient names, 10 profile pairs were not self-matched (i.e., the mRNA and CNV data annotated as having come from the same sample in these cases, were not correlated; see Appendix A). Moreover, we detected six pairs of gene expression and CNV profiles that were cross-matched (i.e., the mRNA profile of one patient significantly correlated to the CNV profile of another patient; see Appendix A). In total, 252 (246 self-matched and six cross-matched) pairs of gene expression and CNV profiles were used in the network reconstruction process.

To identify the source of these sample-labeling errors in the cross-matched mRNA-CNV profile pairs (whether the gene expression or CNV profiles were mis-labeled), the samples’ metadata (clinical annotations) and information inferred from mRNA and CNV profiles were compared. Based on the expression levels of the X-chromosome specific gene XIST and Y-chromosome specific gene RPS4Y1, MMRC samples were clustered into three groups to achieve a clear separation of samples into sex-specific groups (Male Sex, Female Sex, and No Call groups; left panel in Appendix A). There were inconsistencies between annotated sex/gender in the metadata and inferred sex based on the gene expression data for three samples: MMRC0021, MMRC0197, and MMRC0312 (Appendix A). Sex was not inferred for samples in the third group containing profiles in which the levels of expression for both XIST and RPS4Y1 were too low due to X- or Y- chromosome loss in MM cells (left-bottom group in the left panel in Appendix A). Immunoglobin isotype can be inferred based on gene expression profiles. Comparison between the annotated and inferred isotypes revealed one inconsistent sample for light chain isotype (MMRC0207) and two for the heavy chain isotype (MMRC0039 and MMRC0312; Appendix A). Analysis of hyperdiploidy estimated based on CNV profiles [54] and annotated hyperdiploidy yielded one inconsistency for sample MMRC0442 (Appendix A). By comparing the clinically annotated and inferred metadata, the source of labeling errors was unambiguously identified for a few cross-matched profiles. For example, the mRNA and CNV profiles of MMRC0441 and MMRC0442 were cross-matched, and the labeling errors were likely due to a sample swap in CNV profiles based on the consistency with metadata. Similarly, mRNA and CNV profiles of MMRC0312 and MMRC0404 were cross-matched. The sex inconsistency between annotated and inferred sex of MMRC0312 indicated that a sample swap occurred in the mRNA expression profiles (Appendix A). 

After sample labeling error corrections, 304 more cis-associations (in total, 8486) were identified between gene expressions and the respective cis-CNVs at a multiple testing adjusted *p*-value < 0.01. These results confirm that sample-labeling errors frequently occurred in large, complex datasets, and that proMODMatcher can efficiently identify and correct sample-labeling errors to improve power and accuracy in subsequent analyses.

### 2.2. Multiple Myeloma Molecular Causal Network (M3CN) 

CNVs often occur in large blocks, with genes residing in these same blocks, likely sharing common CNV profiles across many samples. As a consequence of the shared common CNVs impacting the expression levels of genes in cis, the expression levels of these genes are likely to be correlated [55,56,57]. For example, the most significant cis-CNV regulated genes (of the 8394 genes identified at false discovery rate (FDR) value < 0.01) were enriched in chr1q (*p*-value = 2.9 × 10^−54^), and these genes were co-expressed (Appendix A). To distinguish gene co-expression due to biological regulations of core molecular and cellular functions from co-expression that is more an artifact of genomic co-localizations to common CNV regions, we included CNV data as nodes in the construction of the molecular causal MM networks, using our previously described Bayesian network reconstruction algorithm [58,59]. Serving as input into the reconstruction algorithm, were 7920 informative genes (mean expression levels > 4.8 and variance of expression levels > 0.4; see Appendix A), of which 3724 were cis-regulated by CNVs (Appendix A). In total 11,644 nodes (7920 nodes for gene expression and 3724 for CNVs) were included in the process of constructing a M3CN using RIMBANet [50] (detailed in Methods). The network reconstruction process searches for a structure G and associated parameters Θ that can best explain the given data P(G,Θ|D), which can be decomposed into a series of substructures (Methods). Given a potential regulation between nodes X and Y (i.e., X and Y are strongly associated), the joint probability p(X,Y|D) can be represented as the structures X→Y p(X,Y|D)=p(Y|X,D)p(X|D), Y→X p(X,Y|D)=p(X|Y,D)p(Y|D), or the structure in which X and Y are both regulated by a third node Z (Figure 2A). Even though there is a directed edge between X and Y, the structures X→Y and Y→X are Markov equivalent (i.e., they have the same probability given the data D, so that they are statistically indistinguishable). However, when cis-CNV nodes are included, serving as a source of perturbation acting on X and Y, the structures X→Y $p\left(X,Y|D\right)=p\left(Y|X,CNV_{y},\;D\right)p(X|D)$ and Y→X $p\left(X,Y|D\right)=p\left(X|Y,\;CNV_{x},D\right)p(Y|D)$ (Figure 2B) are no longer equivalent, so that potential causal relationships between X and Y can be inferred unambiguously. In addition, when conditioning on a given CNV, gene expression correlations due to chromosome co-localization are able to be filtered out. For example, of the 140,283 pairs of (X,Y) that were cis-regulated by CNVs and on the same chromosome associated at a multiple testing adjusted *p* < 0.01, after conditioning on CNV_x (or CNV_y), only 49% of the pairs (X,Y|CNV_x) were associated at the same multiple testing adjusted *p*-value < 0.01, demonstrating the need to integrating CNV data in order to identify true biological regulation.

The resulting M3CN consisted of 9102 interactions between nodes representing variations in gene expression levels. To assess the quality of the constructed network, M3CN was compared with public network or pathway databases that are not MM specific. Nevertheless, a significant number of regulations in M3CN overlapped with interactions in pathway databases, supporting the biological relevance of the network. For example, over 30% of the regulations captured by the M3CN overlapped with regulatory relationships represented in the KEGG and Hallmark databases (Appendix A), indicating that M3CN was recapitulating known biological pathways. Causal links in M3CN were also compared with top correlated gene pairs derived from more basic co-expression analysis. Of the top 9102 gene pairs with the highest Pearson’s correlation coefficients between their expression profiles, the correlations were more often being driven by genomic co-localization rather than coherent regulation of biological processes (Appendix A). 

The M3CN revealed many known features of MM. For example, the IgH gene [4] is one of the most frequently translocated genes in MM. The subnetwork associated with IgH is composed of mostly IgH family genes or genes located in the same region as IgH (14q32.33). However, this subnetwork is almost completely isolated from the rest of M3CN, with only a single node (the gene MIR8071-2) connecting this subnetwork to the rest of M3CN, (Appendix A This suggests that the pattern of gene expression variation driven by IgH and its corresponding subnetwork may have a limited impact on the rest of M3CN. Translocations between the immunoglobin heavy chain locus and oncogene loci, including *CCND1*, *CCND3*, *MAF*, *FGFR3*, and *MMSET* (*WHSC1*) commonly occur in MM patients [60]. Further, MM patients can be divided into TC1-5 molecular subtypes based on translocations and *CCND1-3* expression levels [61]. Subtype-specific signatures were derived based on GSE13591 [62]. Putative key regulators were inferred for the TC subtype-specific signatures, including *CCDN1* and *WHSC1* as key regulators for the TC1 and TC4 subtype-specific signatures, respectively. TC1-3 subtypes had one of the D-cyclin genes, *CCND1-3*, highly expressed, and the CD4/6-Rb pathway [63] activated. The subnetworks for the TC1-3 subtype-specific signatures all significantly overlapped with each other (e.g., the subnetwork for TC1 overlapped the TC2 and TC3 subnetworks, with TC1 2.2-fold enriched for TC2, *p*-values = 5.7 × 10^−25^, and 6.2-fold enriched for TC3, *p*-value = 1.6 × 10^−39^), consistent with the observations that MM patients of TC1-3 subtypes shared similar survival patterns [64]. In contrast, *WHSC1* (*MMSET*) was upregulated in the TC4 subtype; *WHSC1* regulates the histone methylation of MM cells [65], which in turn regulates cell proliferation. The subnetwork for the TC4 specific signature was distinct from the subnetworks for the TC1-3 specific signatures (overlaps were not significant), consistent with the observations that MM patients of TC4 subtype had worse prognosis than the ones of TC1-3 subtypes [64].

At the global level, there were two highly connected genes, AGPS (Alkylglycerone Phosphate Synthase) and ATRX (Alpha Thalassemia/Mental Retardation Syndrome, X-Linked), regulated dozens of genes directly (41 and 32 respectively, Appendix A). AGPS is a metabolic enzyme, a critical component in the synthesis of ether lipids, and is up-regulated across multiple types of aggressive human cancer cells and primary tumors [66]. Multiple studies show that lipid metabolism plays a critical role in MM tumorigenesis and progression [67]. Previous studies have also shown the potential of AGPS as a therapeutic target of cancer, and multiple AGPS inhibitors are in development [68]. ATRX is a chromatin remodeling protein whose main function is the deposition of the histone variant H3.3. A recent study showed that ATRX is a potential mutational driver in MM [69]. 

### 2.3. MM Prognostic Signature Genes in the M3CN

Eight large prognostic gene expression signatures were collected from the literature, with the number of genes across these eight signatures ranging from 15 to 92 (Table 1). The overlap of genes among these different signatures was limited (Appendix A). For example, only one gene, *BIRC5*, appeared in more than two signatures (Hose_50, Shaughnessy_70 and Kuiper_92 (EMC-92)). Some of the signatures were enriched in specific chromosomal locations (Appendix A). For instance, 14 genes from Shaughnessy_70 and 13 genes from Burington_92 were located in chromosome 1 (Fisher’s exact test *p* = 5.54 × 10^−5^ and 2.70 × 10^−2^, respectively), consistent with the frequently described chromosome 1 aberration in MM [70].

These prognostic gene signatures were projected onto M3CN to infer the key regulators of each signature. It is of note that only a fraction of genes in each signature were included in the M3CN (Table 1), as only the most informative genes (e.g., those genes detectably expressed and that were observed to vary across samples) were included in the M3CN construction procedure (detailed in Methods). Our network analysis identified 15 potential key regulators for three of the eight prognostic signatures (Appendix A, detailed in Methods), including *NOP16* and *CECR5* for Shaughnessy_70; *MELK*, *TPX2,* and *NCAPG2* for Kuiper_92; and *CDK1*, *DTL,* and eight other genes for the Hose_50 signature. *TPX2* is known to regulate *AURKA* and interacts with *RHAMM* [71], which is known to correlate with centrosome amplification and with poor prognosis in MM [72]; the *AURKA* inhibitor is a potential treatment for MM [73]. *CDK1* is one of the key regulators, and its inhibition has potential as an anti-cancer treatment [74]. Cyclin-dependent kinase inhibitors have been shown to induce cell cycle arrest and eventual apoptotic cell death in MM cells [75].

Using the 15 putative key regulators that we identified as regulating the prognostic gene signatures, we extracted a subnetwork consisting of 178 nodes and 218 interactions (Appendix A) from M3CN (Figure 3, detailed in Methods) with the 15 key regulators as seeds, referred to as the prognostic subnetwork hereafter. Even though different prognostic signatures overlapped with each other sparsely at the individual gene level (Appendix A), five of the eight prognostic signatures we examined, were significantly enriched in this subnetwork (Table 1). For example, 28, 16, and 14 genes in Hose_50, Kuiper_92, and Shaughnessy_70, respectively, were in the prognostic subnetwork (corresponding to Fisher’s exact test *p*-values: 2.20 × 10^−16^, 2.10 × 10^−13^, and 1.45 × 10^−14^, respectively). Other smaller signatures, such as Kassambara_22 and Reme_19, were also enriched in this subnetwork (*p* = 4.80 × 10^−7^ and 1.55 × 10^−7^, respectively). 

The prognostic subnetwork was enriched for genes involved in the cell cycle and metabolic processes (Appendix A, detailed in Methods). More specifically, 42 out of 178 genes are involved in the cell cycle (fold enrichment = 8.4, FDR = 1.1 × 10^−22^), and 25 in mitotic cell cycle process (FDR = 7.81 × 10^−18^). Other top pathways include DNA replication (20 genes, FDR = 1.42 × 10^−15^), cellular macromolecule biosynthetic process (20 genes, FDR: 8.8 × 10^−5^), and cellular response to DNA damage stimulus (16 genes, FDR: 1.75 × 10^−8^). The results are consistent with the essential roles cell cycle genes are known to play in MM progression, and the therapeutic implications [76].

To assess the prognostic values of the prognostic subnetwork or other prognostic signatures in the literature, we applied them to the Multiple Myeloma Research Foundation, MMRF’s, CoMMpass cohort (Methods), which was not used in constructing M3CN or training other prognostic signatures. In addition to the above prognostic signatures, we also included a four-gene signature [48] that was developed based on the MMRF-CoMMpass RNAseq dataset for comparison. Hierarchical clustering was applied to the CoMMpass RNAseq data based on genes in each signature, and the clustering result suggested that MM patients could be divided into three groups in general (Figure 4A and Appendix A). Then, k-means clustering (k = 3) was applied to the data (Figure 4B and Appendix A), and the three groups were compared in terms of overall survival (OS) (Figure 4C and Figure 5) and progression free survival (PFS) (Figure 4D and Figure 6). The patient groups, based on the prognostic subnetwork, had significantly different survival (log-rank test *p*-values = 1.8 × 10^−12^ and 1.7 × 10^−10^), and hazard ratios (HRs) between high and low risk groups (4.7 and 3.3 for OS and PFS, respectively), were the best among all signatures tested (Table 2).

### 2.4. MM Treatment Response Signature Genes in the M3CN Network

To explore treatment response signatures in the context of the M3CN network, we identified five large signature gene sets (number of genes ≥ 15) related to various aspects of treatment in MM from the literature (Table 3). These signature genes encompass common types of treatments for MM. They also capture different aspects of treatments, including signature genes associated with patient survival after certain treatments (i.e., an after-treatment survival signature, such as shown in Mulligan_100), genes associated with drug resistance in MM cells (resistance signature, Mitra_42), genes whose expression changed after treatment (treatment effect signature, Bhutani_176), and genes identified through drug binding affinity screen (drug mechanism signature, Zhu_244). Similar to the prognostic gene signatures, the overlap among the drug-response signatures was limited: no genes were shared by three or more publications and only 13 genes were shared by two publications: *FLNA*, *SSX4*, *FAIM3*, *ITGB7*, *PPP1R16B*, *HLA-DOB*, *DEK*, *BLVRB*, *FKBP5*, *HCLS1*, *IDH1*, *DNAJA1*, and *XPO1*. 

#### 2.4.1. Immunomodulatory Drugs (IMiDs) Response Signatures

The mechanisms of IMiDs are complex, including apoptosis, immune responses, and anti-angiogenesis [77]. Two response signatures, which were derived from completely different approaches, were reported in the literature (Table 2) [21,42]. Zhu et al. reported 244 proteins (Zhu_244) that bind to E3 ligase protein cereblon (*CRBN*), which is the target of IMiDs [42], while Bhutani et al. report 176 genes (Bhutani_176) that were differently expressed when comparing bone marrow samples before and after IMiD treatment [21]. The two IMiDs’ signatures were enriched for genes in different biological processes (Appendix A), indicating the two studies captured different aspects of IMiD treatment. 

Zhu_244 was enriched for genes on Chromosome 4 (*p*-value = 0.007; Appendix A). Among the 244 genes in this signature, 143 were included in the M3CN, and 16 of them were in a subnetwork of 117 nodes (the largest connected subgraph and noted as subnetwork_Zhu_244, *p* = 2.3 × 10^−10^; Appendix A). Subnetwork_Zhu_244 was enriched for the organic substance metabolic process (*p* = 4.75 × 10^−5^; Appendix A). One of the key regulators of the subnetwork_Zhu_244 was *CCT2*, a chaperonin containing TCP1 subunit 2 (Appendix A). *CCT2* is also in the prognostic signature Shaughnessy_70 [38]. In the M3CN, *CCT2* regulated *ADSL* (Appendix A), with both of these genes also included in the prognostic subnetwork (Figure 3). The subnetwork Zhu_244 and the prognostic subnetwork significantly overlap (*p* = 2.9 × 10^−4^), suggesting IMiDs bind *CRBN* and in turn suppressing cell proliferation [77]. 

On the other hand, 132 of 176 genes in Bhutani_176 were included in the M3CN, and 19 were in a subnetwork comprised of 79 nodes (the largest connected subgraph and noted as subnetwork_Bhutani_176, *p* = 2.82 × 10^−17^; Appendix A). *MX1*, an interferon-regulated resistance GTP-binding protein, was one of the key regulators of subnetwork_Bhutani_176 (Appendix A). *MX1* is a key player of MM defined in proteomic studies [78]. Five genes (*MX1*, *STAT1*, *IFI27*, *IFI35,* and *OAS1*) in the *MX1*-regulated subnetwork overlapped with the 121 genes regulated by the DNA methylation inhibitor decitabine in MM cells; three of them (*IFI27*, *IFI35,* and *OAS1*) were also directly connected with *MX1* in subnetwork_Bhutani_176 [79]. *SOCS3*, suppressor of cytokine signaling 3, was also a key regulator of subnetwork_Bhutani_176 (Appendix A). SOCS3, together with STAT3 regulate the proliferation of MM cells [80,81]. The subnetwork_Bhutani_176 did not overlap with the prognostic subnetwork, and only shared one gene, *PRB1*, with subnetwork_Zhu_244 (Appendix A), suggesting pleiotropic effects of IMiDs.

Furthermore, we generated a subnetwork by combining the Zhu_244 and Bhutani_176 signatures (Figure 7), which has 239 genes (Appendix A) and 244 edges (Appendix A), including 50 genes from Zhu_244 and 12 genes from Bhutanis_176. Two genes in the subnetwork, *XPO1* and *IDH1* were present in both signatures; seven edges connect two signatures. Pathway analysis showed that the subnetwork enriched genes as a defensive response to viruses (*p* = 2.47 × 10^−9^) and metabolic processes (*p* = 2.76 × 10^−6^) (Appendix A).

#### 2.4.2. Proteasome Inhibitor (PI) Response Signatures

Mulligan et al. report a signature of 100 genes that associated with the survival of patients treated by proteasome inhibitor bortezomib [33]. Among them, 10 genes (including *GAGE4*, *MAGEA3*, *MAGEA6*, and *SSX2*) were on X chromosome (Appendix A; *p* = 2.9 × 10^−5^). Melanoma Antigen genes (MAGEs), more specifically *MAGEA3*, promote the survival of MM cells [82]. Cancer-testis antigen genes (*MAGEA3, MAGEA6,* and *GAGE4*) co-express with cell cycle genes (e.g., *CCNB2* and *MCM2*) in high proliferation MM patients [64], suggesting that PI drugs work better in highly proliferative MM cells [83]. Thirty-four genes in the signature were included in the M3CN, and the subnetworks of the signature (noted as subnetwork_Mulligan_100) consisted of 48 genes (Appendix A), including nine genes in the signature (*p* = 2.3 × 10^−13^). *DEFA1*, defensin alpha 1, is significantly downregulated in MM [84], but upregulated in the Myeloid subgroup of MM patients [85]. MM patients with high myeloid signatures had much better survival [86]. Note that *MAGEA3*, *MAGEA6*, and *GAGE4* have positive weight in Mulligan_100, whereas *DEFA1* has a negative weight in the original publication [33].

Mitra_42 (Table 3) is derived by treating myeloma cell lines with multiple PIs [32]. This in vitro signature was not enriched in the M3CN and did not overlap with the Mulligan_100 signature, suggesting the in vitro signature does not reflect regulations in vivo. 

#### 2.4.3. Drug-Combination Response Signatures

The Shaughnessy_80 signature (Table 3) was derived in a two-step way: first, identifying differently expressed genes by comparing before and after thalidomide and bortezomib; then, differentially expressed genes (DEGs) are filtered based on association with survival [37]. Forty genes in the signatures were included in the M3CN. The subnetwork of Shauhnessy_80 consisted of 41 genes with eight genes in the signature (*p* = 1.7 × 10^−11^; Appendix A). There were five genes in the overlap between subnetwork_Shauhnessy_80 and the IMiD response signature subnetwork, subnetwork_Zhu_244 (*p* = 3.2 × 10^−4^; Supplementary Materilas Appendix A), but there was no overlap between subnetwork_Shauhnessy_80 and subnet_Bhutani_176, nor the former and subnet_Mulligan_100.

## 3. Discussion

Our network model integrated the multi-omics data available on MM patients. During data preprocessing for multi-omics data integration, our analyses suggested that samples of 12 patients in the MMRC dataset (5% of the total number of patients) were annotated incorrectly due to sample swaps and other unknown causes. Sample-labeling errors occur in clinics [87], clinical trials [88], and research databases [52]. Analyses based on error-containing datasets might decrease the power of the datasets, but may also lead to incorrect or contradictory scientific conclusions, or even harming patients [89,90]. Thus, it is critical to perform extensive QC to identify and correct potential sample-labeling errors in large-scale, multi-omics datasets prior to integrative data analysis, especially in the field of precision medicine. 

Our network model aimed to identify molecular causal regulatory relationships between genes and among gene groups, with the ability to distinguish between these relationships and those co-expression patterns that may be driven by both biological regulation and genomic co-localization. A co-expression network of MM was constructed based on the MMRF-CoMMpass data by Lagana et al. [48]. The M9 module in Lagana et al.’s network was associated with prognosis (*p* = 5 × 10^−3^), and enriched for genes in the 13q deletion and 1q amplification blocks [48]. There were 972 unique genes in the M9 module, and 120 of them overlapped with our prognostic subnetwork (fold enrichment =14.1, *p* = 7.9 × 10^−41^). Even though they was significantly overlapping, genes in our prognostic subnetwork were not biased with respect to any chromosome location. The expressions of genes in our prognostic subnetwork were used to cluster patients in the MMRF-CoMMpass study into three risk groups (Figure 4B); these groups were significantly different in terms of overall survival and progression free survival (log-rank test *p* = 1.8 × 10^−12^ and 1.7 × 10^−10^, respectively), with hazard ratios (HRs) of 4.7 and 3.3, which were larger than the corresponding HRs based on the M9 module as a whole (HR = 1.75 for PFS), and the best four-gene combination used to predict prognosis (HR = 3 for PFS as reported in Lagana et al.). This is of particular note, given that the MMRF-CoMMpass study was not used in constructing the prognostic subnetwork, while Lagana et al.’s network was built on the MMRF-CoMMpass dataset. The results further suggest that our prognostic subnetwork is robust in identifying molecular mechanisms underlying high progression risk in MM patients.

With the increase of therapeutic options for MM patients, the development of an efficient prognostic and predictive signature becomes important, as personalized treatment plans can be tailored for patients with different risks, so that both under and over-treatment can be avoided. Molecular factors, such as genetic aberrations and gene expression profiles, have been investigated for their prognostic/predictive ability. Several genetic aberrations have been added to the Revised International Staging System (R-ISS), such as del(17p), t(4;14) and t(14;16), which are consistently associated with poor survival. On the other hand, studies of gene expression signatures have not led to consistent results, as shown by our analysis, where the number of genes in each signature varied significantly and the overlap among signatures was extremely limited. There are several potential reasons for those inconsistencies. First, the heterogeneity of MM may contribute to the inconsistencies. Though MM is still considered a single disease entity, its clinical presentation, response to treatment, and survival outcomes are highly heterogeneous. Based on cytogenetic abnormalities, MM can originate from two pathways: the hyperdiploidy of chromosomes 3, 5, 7, 9, 11, 15, 19 and 21, which is observed in 55% of patients; the translocation of *IGH* or *MYC*, observed in 40% to 50% of patients [4]. In most cases, hyperdiploidy and translocation are mutually exclusive in MM patients. The clustering analysis of genomic landscape and/or gene expression profiles reveal four to 10 subgroups using a large number of MM patients [44,69], indicating the underlying biology of these two types of MM is different. It is very likely that signatures will be different for those subgroups, but so far there are no signatures developed for subgroups separately. Second, variability in the methods applied to derive the signatures may result in differences among the signatures. Although almost all signatures were derived from microarray-based gene expression profiling, the methods are very different. For example, Kuiper_92 was developed using univariate Cox regression analysis, followed by a supervised principal component analysis in combination with simulated annealing [31], whereas Shaughnessy_70 was derived by a combination of log-rank tests with a stepwise multiple linear discriminant analysis [38], and then the method of Shaughnessy_80 used the expression profile after 48-h bortezomib test-dosing [37]. More recently, a simulation-based sophisticated machine learning approach was applied to derive a signature to predict the treatment benefits for MM patients [91]. Given the big differences among the applied methods, it is not surprise that the overlap among signatures was very limited. 

To develop personalized treatments, it is critical to separate prognosis without a specific treatment and with the benefit due to the specific treatment. Even though the three prognostic signatures (Hose_50, Kuiper_92, and Shaugnessy_70) overlapped with each other sparely, they were all enriched in the prognostic subnetwork (Figure 2), indicating that they were closely connected and regulated, and potentially in the same or related biological pathways; our pathway analysis confirmed that the prognostic subnetwork enriched genes of the cell cycle process (Appendix A). The prognostic subnetwork developed in this study reflects most prognostic signatures and has the potential to enable more robust estimation of baseline prognosis (Figure 4, Figure 5 and Figure 6 and Table 2) when developing new treatments for MM.

Our network analysis also showed that MM drugs had pleiotropic effects with signature genes enriched in different subnetworks. However, some MM signatures were not enriched in any MM subnetwork, such as Mitra_42. The potential reasons for that disconnection may be due to (1) MM heterogeneity as discussed above, and (2) MM cells’ interactions with the tumor microenvironment. Recent studies showed that bone marrow niche is critical for MM cell survival and proliferation [92]. Bone marrow adipocytes provide metabolites and metabolic signals for MM cell proliferation [67]. Thus, in vitro signatures may not faithfully reflect changes in vivo. As new high-throughput technology, such as single cell RNA sequencing [93], rapidly develops and is applied in studying diseases, including MM [94,95], more understanding of the MM microenvironment and treatment response is expected in the near future. 

## 4. Materials and Methods 

### 4.1. The Preprocess of Gene Expression, and CNV Data and Omics Data Matching 

We downloaded the MMRC reference dataset consisting of 304 mRNA expression profiles (GSE26760), 254 CNV profiles (GSE26849), and associated metadata [49]. For the CNV profiles, the downloaded Circular binary segmentation (CBS) of the logR ratio values was mapped to gene levels’ CNVs based on the coordinate information from Refgene annotations against hg19. The coordinate was downloaded from UCSC genome browser (https://genome/ucsc.edu). The sex prediction of MM patients was based on the expression levels of X chromosome specific gene *XIST* and Y chromosome specific gene *RPS4Y1* [96]. Immunoglobulin light chain and heavy chain isotypes of each MM sample were predicted by the corresponding Immunoglobulin genes. CNV data was used to estimate hyperdiploidy using a previous published method [97].

*pro*MODMatcher was applied to detect sample labelling errors using mRNA and CNV profiles; the detail of the method can be found in [51]. Briefly, starting with 246 mRNA and CNV pairs that were matched based on their labels, we first identified a set of genes whose mRNA expression was highly correlated with their CNV profiles across all the 246 sample pairs. Out of 20,395 genes, 8404 genes were significantly correlated between mRNA and CNV profiles (FDR < 0.01). Then the profiles of selected genes were rank-transformed for non-parametric comparison among samples. Finally, the sample similarity score was calculated as the correlation coefficient between RNA expression of cis genes in sample i and the CNV values of cis genes in samples j, $S_{g_{\mathrm{cnv}}}\left(i,j\right)$ for all i, j=1…n. We expected that (1) $S_{g_{\mathrm{cnv}}}\left(i,i\right)$ was greater than $S_{g_{\mathrm{cnv}}}\left(i,j\right)$ for j≠i, and (2) $S_{g_{\mathrm{cnv}}}\left(i,i\right)$ was greater than $S_{g_{\mathrm{cnv}}}\left(k,i\right)$ for ≠i, as one sample mRNA profile of cis genes should be the most similar to its own CNV profile and vice versa. If neither (1) nor (2) was satisfied for one sample, we tried to identify potential cross matches, such that $S_{g_{\mathrm{cnv}}}\left(i,j\right)$ was the most significant among all possible permutations. With updated gene expression and CNV profiles pairs, cis gene expression-CNV pairs were refined and the procedure was iterated until there was no update between iterations. 

### 4.2. Gene Signatures for MM

Twenty-two gene signatures for MM were collected from publications up to April 2019, 13 of them with numbers of genes > 15 were used for network analysis and were listed in Table 1 and Table 2, along with the PMID of the publication. Probe IDs of microarrays were converted to gene symbols based on annotation of the relevant microarray platform. Duplicated gene symbols were removed, as were probes that failed to map to any annotated genes. For signatures that are not lists of genes, for example, the ratio of two genes, the resulting signature contained both genes.

### 4.3. M3CN Construction and Network Analysis

The M3CN was constructed by integrating gene expression and CNV profiles. GSE26863 (GSE26760 and GSE26849) for MM patients was downloaded from GEO, processed and normalized using R/Bioconductor. In total, 7920 genes with detectable expression levels and large variances across samples were selected to be included in the network reconstruction process. Among them, the expression of 3724 genes was cis-regulated by CNVs (FDR < 0.01) so that 3724 nodes for cis-CNVs were also included in the network reconstruction. The gene expression or CNV profile was discretized into three states: low, normal, and high level, guided by k-means clustering (k = 3) and biological meaningful cutoff values [98]. The gene expression and CNV nodes were then input into the software suite, Reconstructing Integrative Molecular Bayesian Network (RIMBANet) [50,99], to construct a biological causal network given the data and priors, as previously described. Briefly, the network reconstruction process searches for a directed acyclic graph (DAG) structure G and associated parameters Θ that can best explain the given data D, P(G,Θ|D). If the structure G is a DAG, then P(G,Θ|D) can be decomposed into a series of sub-structures $P\left(G,\Theta |D\right)=\underset{i}{\prod }P(G^{i},\;\Theta^{i}|D)$. With cis-CNV nodes included (Figure 2B), the structures X→Y, given by $\;p\left(X\to Y|D\right)=p\left(Y|X,CNV_{y},\;D\right)p(X|D)$, and Y→X, given by $p\left(Y\to X|D\right)=p\left(X|Y,CNV_{x},D\right)p(Y|D)$, are no longer equivalent, so that potential causal relationships between X and Y can be inferred unambiguously. To speed up the searching process, for each gene, the bottom 20% genes based on their mutual information were excluded as potential candidate regulators (sparse candidate search [100]). The network reconstruction process is a Monte Carlo Markov chain (MCMC). Given different random seeds, we might end up with different structures. Thus, we ran 1000 independent MCMC processes based on 1000 random seed numbers that resulted in 1000 candidate structures. Then, we selected consensus structure features with posterior probabilities >0.3 among candidate structures [98]. Finally, loops in the consensus network were removed by deleting the weakest link in the loops. The resulting network was visualized using Cytoscape 3.7 [101]. Given a set of seed nodes $N_{s}$, $N_{s}=\underset{i}{⋃}d(node,\;N_{s}^{i})\leq l)$ is the union of nodes that are within l steps from the seed node $N_{s}^{i}$, and the subnetwork for the seed nodes $N_{s}$ is the set of connections among $N_{s}$.

### 4.4. The Identification of Key Regulators for Signature Genes

For the three signature genes, Hose_50, Huiper_92, and Shaugnessy_70, we further identified the putative key regulators [50,102,103] in M3CN. Briefly, for each gene *i* in the M3CN, a two-step subnetwork sub*_i_* (*l* = 2) was extracted as described above. If sub*_i_* was enriched for the input signature genes, then the gene *i* was a candidate regulator. After all candidate regulators were identified, they were sorted by corresponding enrichment *p*-values. The gene with the most significant enrichment *p*-value was a key regulator. Then, any candidate regulators in its two-step subnetwork were excluded as candidate key regulators. Moreover, the next candidate regulator in the sorted list was selected as a key regulator. The process was repeated throughout the sorted candidate list. 

### 4.5. Pathway Analysis

PANTHER was used for pathway analysis based on overrepresentation test [104]. The gene list was submitted to PANTHER web server (version 14.1); Fisher’s exact test was used for statistical hypothesis testing. FDR was used for the correction of the multiple-hypothesis testing problem, with FDR < 0.05 as the cutoff. 

### 4.6. The MMRF-CoMMpass Cohort

The IA14 Release of CoMMpass data was downloaded from the MMRF researcher gateway portal (https://research.themmrf.org), which consists of 767 RNA sequencing data of baseline CD138+ plasma cell bone marrow samples from newly diagnosed MM patients. Among them, overall survival or progression-free survival information was available for 648 patients which were included in our analyses. 

## 5. Conclusions

We constructed a molecular causal network describing potential regulatory relationships in MM and analyzed different MM gene signatures using the network. We identified a prognostic subnetwork that shared among different MM prognostic signatures and showed that the prognostic subnetwork was more robust than individual signatures in terms predicting prognosis in an independent cohort. Our study demonstrated that MM was complex so that an integrative network is needed to reveal molecular mechanisms underlying different MM signatures.

## Figures and Tables

**Figure 1 cancers-11-01452-f001:**
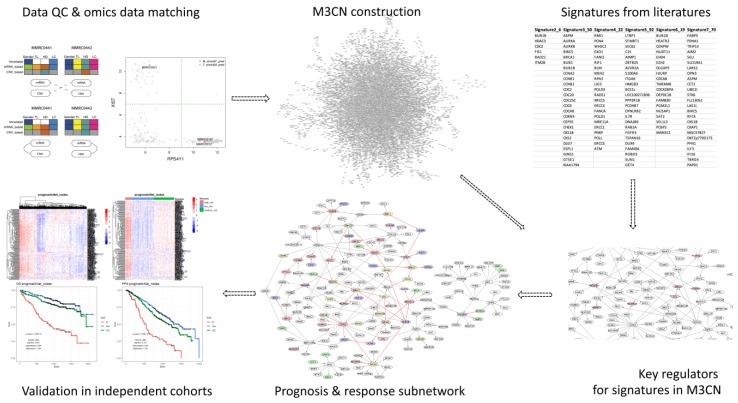
Workflow in this study. After extensive QC and omics data matching to correct sample-labeling errors, multiple myeloma molecular causal network (M3CN) was constructed, and multiple myeloma (MM) prognostic signature gene sets from the literature were projected onto it. Network analysis was subsequently performed to identify subnetwork associated with prognosis/response. The predictive power of subnetworks was assessed in independent cohorts.

**Figure 2 cancers-11-01452-f002:**
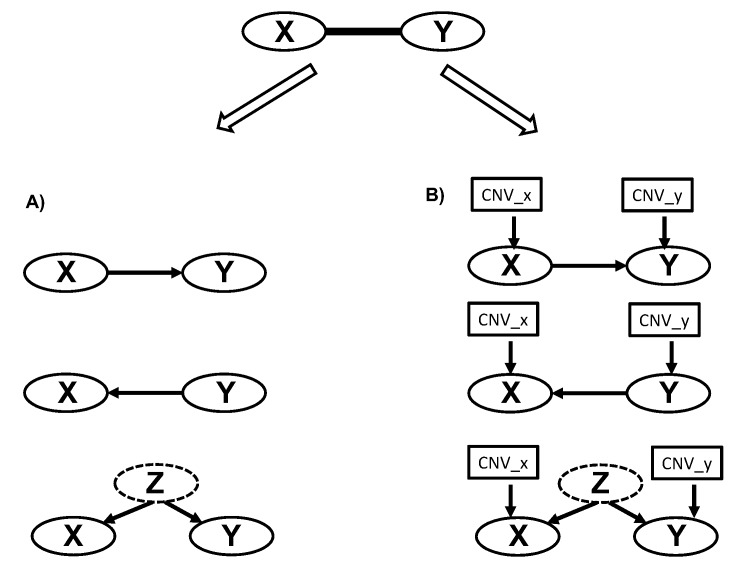
The causal structural inference schemes based on gene expression data, (**A**) without copy number variation (CNV) and (**B**) with CNV data integrated.

**Figure 3 cancers-11-01452-f003:**
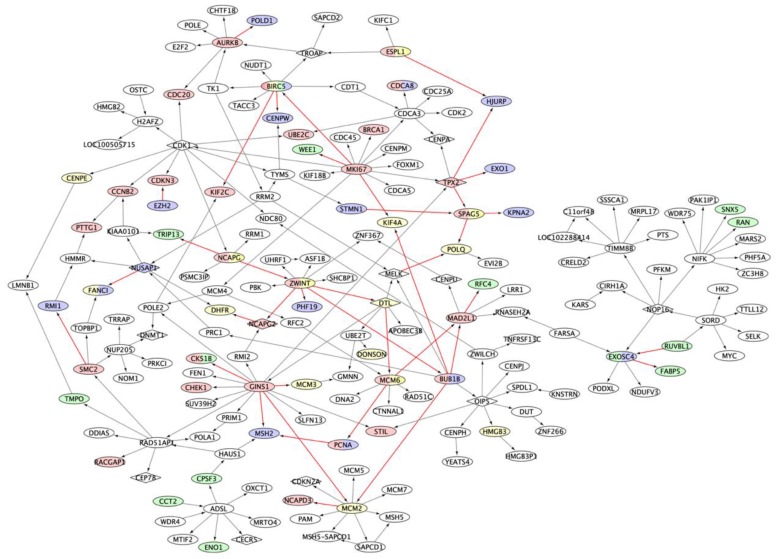
The prognostic subnetwork. The subnetwork was generated using key regulators for the Hose_40, Kuiper_92, and Shaughnessy_70 signatures as seeds. Nodes are color coded based on signatures: yellow nodes are genes from Kuiper_92, green nodes from Shaughnessy_70, and red nodes are Hose_40 genes; blue nodes are genes of other signatures. Edges in red indicate connections between genes of different signatures. Nodes of a diamond shape indicate key regulators for signatures.

**Figure 4 cancers-11-01452-f004:**
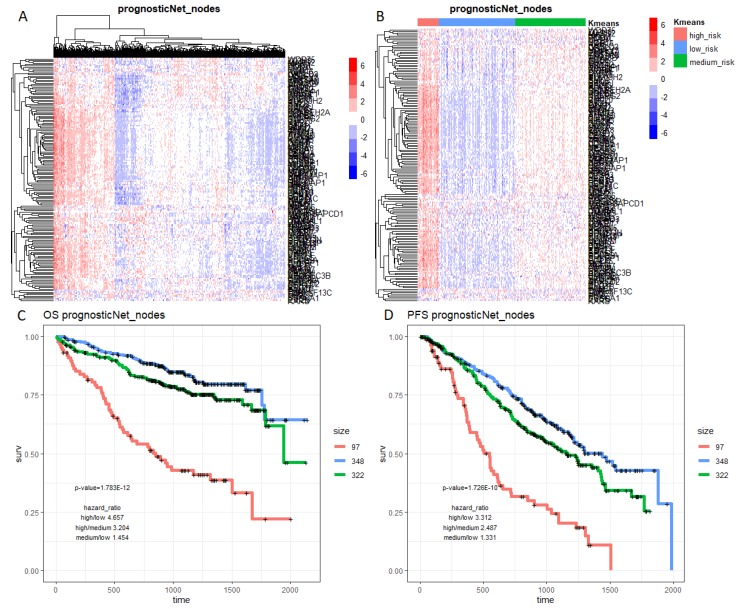
Heatmaps and K–M plots based on genes in the prognostic subnetwork and CoMMpass data. Rows in the heatmaps are genes and columns are samples. (**A**) The hierarchical clustering result and (**B**) the k-means (k = 3) clustering result were based on the z-scores of expression levels. K-M plots of three patient groups (based on k-means clusters) for (**C**) overall survival (OS) and (**D**) progress free survival (PFS).

**Figure 5 cancers-11-01452-f005:**
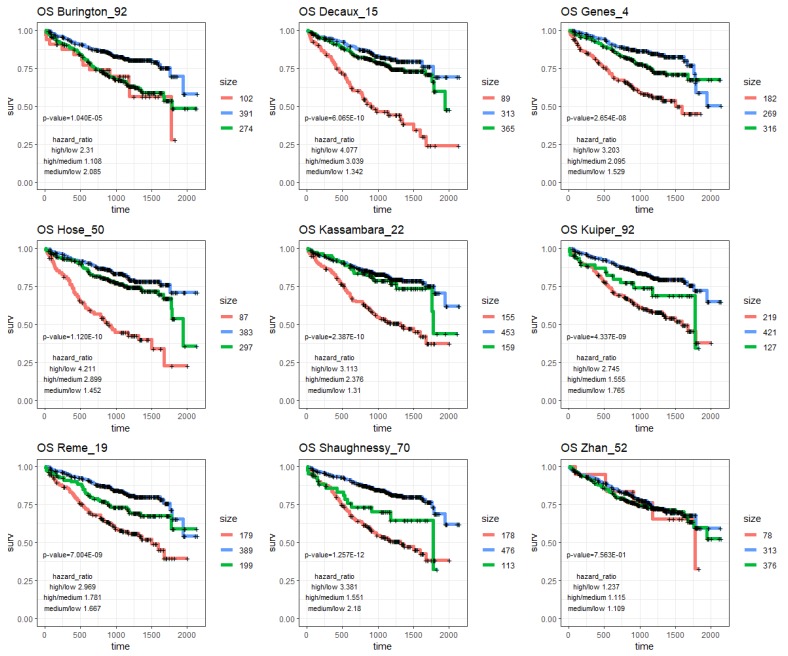
K-M plots of OS in patients in the CoMMpass study. Patients were partitioned into three groups based on genes in the nine different prognostic signatures in the literature (Appendix A).

**Figure 6 cancers-11-01452-f006:**
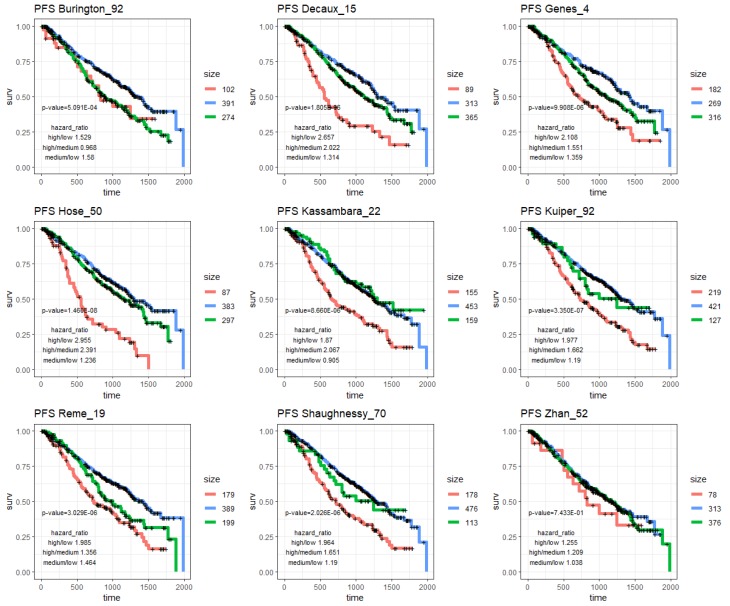
K-M plots of PFS in patients in the CoMMpass study. Patients were partitioned into three groups based on genes in the nine different prognostic signatures in the literature (Appendix A).

**Figure 7 cancers-11-01452-f007:**
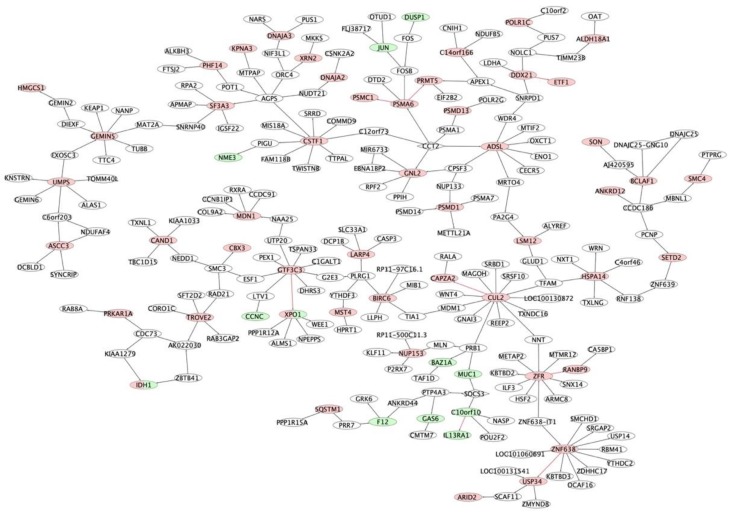
The IMiDs’ treatment response subnetwork. The subnetwork was generated using a combination of Zhu_244 and Bhutani_176 signatures. Nodes are color coded based on signatures: red nodes are genes from Zhu_244, and green nodes from Bhutani_176 genes. Edges in red indicate connections between genes of signatures. Nodes which are diamond shaped indicate key regulators for signatures.

**Table 1 cancers-11-01452-t001:** MM prognostic signature genes collected from literature.

Signature	PMID	Num. of sig. Genes	Num. of Genes in M3CN	Num. of Genes in Subnet	*p*-Values
**Burington_92**	18676754	92	62	0	0.41
**Decaux_15**	18591550	15	6	1	0.13
**Hose_50**	20884712	50	30	28	2.2 × 10^−16^
**Kassambara_22**	24809299	22	14	6	4.8 × 10^−7^
**Kuiper_92**	22722715	92	55	16	2.1 × 10^−13^
**Reme_19**	23493321	19	12	6	1.6 × 10^−7^
**Shaughnessy_70**	17105813	70	33	14	1.5 × 10^−14^
**Zhan_52**	17023574	52	31	0	1.0

**Table 2 cancers-11-01452-t002:** Summary of differences among MM patients in different risk groups, stratified based on genes in the prognostic subnetwork and the prognostic signatures in the literature. A total of 648 patients with survival information in the MMRF-CoMMpass study were included in the analysis. *p*-values were log-rank test *p*-values based on three risk groups. HR: hazard ratio.

Signature	OS				PFS			
	*p*-Value	HR High/Low	HR High/Med.	HR Med./Low	*p*-Value	HR High/Low	HR High/Med.	HR Med./Low
progNet	1.78 × 10^−12^	**4.657**	**3.204**	1.454	**1.73 × 10^−10^**	**3.312**	**2.487**	1.331
Burington_92	1.04 × 10^−5^	2.31	1.108	2.085	5.09 × 10^−4^	1.529	0.968	1.58
Decaux_15	6.07 × 10^−10^	4.077	3.039	1.342	1.81 × 10^−6^	2.657	2.022	1.314
Genes_4	2.65 × 10^−8^	3.203	2.095	1.529	9.91 × 10^−6^	2.108	1.551	**1.359**
Hose_50	1.12 × 10^−10^	4.211	2.899	1.452	1.46 × 10^−8^	2.955	2.391	1.236
Kassambara_22	2.39 × 10^−10^	3.113	2.376	1.31	8.66 × 10^−6^	1.87	2.067	0.905
Kuiper_92	4.34 × 10^−9^	2.745	1.555	1.765	3.35 × 10^−7^	1.977	1.662	1.19
Reme_19	7.00 × 10^−9^	2.969	1.781	1.667	3.03 × 10^−6^	1.985	1.356	1.464
Shaughnessy_70	**1.26 × 10^−12^**	3.381	1.551	**2.18**	2.03 × 10^−6^	1.964	1.651	1.19
Zhan_52	7.56 × 10^−1^	1.237	1.115	1.109	7.43 × 10^−1^	1.255	1.209	1.038

**Table 3 cancers-11-01452-t003:** MM drug-response signatures collected from the literature.

Signature	PMID	Num. of Sig. Genes	Num. Genes in M3CN	Treatment	Patients
**Bhutani_176**	28863804	176	132	IMiDs	Mixed
**Zhu_244**	24914135	244	143	IMiDs	Mixed
**Mitra_42**	28665416	42	29	PI	Mixed
**Mulligan_100**	17185464	100	30	PI	Mixed
**Shaughnessy_80**	21628408	80	40	TT2/TT3	NDMM

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
