# Peer review of "A Network Analysis of Multiple Myeloma Related Gene Signatures"

_cancers, 2019, doi:10.3390/cancers11101452_

Round 1

Reviewer 1 Report

Liu et al. describe the generation of a Molecular Causal Network based on the correlation of gene expression and copy number variations in the MMRC multiple myeloma dataset. In addition, the network was used to generate "prognostic" and "treatment response" subnetworks based on existing publically available gene signatures. The general idea is interesting but not so novel as recent work using the MMRF-COMPASS study already provided an extensive network analysis (Lagana et al, Leukemia, 2017). Several comments should be adressed to improve the quality of the manuscript.

Major comments:

The description of the generation of M3CN network is not so transparant. This should be supplemented with statistical data showing the actual list of genes, CNVs, correlation etc... in supplemental tables.

The same comment applies to the description of the "prognostic" and "treatment response" signatures and subnetworks. The authors should provide additional tables containing gene lists and statistical values to support the evidence that the subnetworks provide. For example: 15 key regulatory genes were extracted but no data is provided that explains where the p-value comes from that selected these 15 genes. Likewise, the actual genes that contribute to the prognostic subnetwork and treatment response signatures should be provided in tables together with statistical values to provide evidence for their presence in the M3CN or subnetwork.

What is still missing is the proof that the obtained networks or signatures can be actually be used to predict survival of multiple myeloma patients of the MMRC cohort and why this network is better in comparison to the already existing signatures. The network should also be validated in an independent cohort such as the MMRF-COMPASS cohort. In terms of pathways, cell cycle is already known to be a very prognostic factor for survival of myeloma patients and therefore does not provide novel findings. The authors should stress more on the potential novel implications or findings of their analysis. How can their information be used to stratify patients for individualized therapy? Can the network be used to actually predict how a patient will respond to certain treatments? This should be validated in other cohorts or at least discussed how this is practically possible.

Minor comments:

In the introduction, the recent approval of monoclonal antibodies such as Daratumumab should be mentioned as one of the newest therapies.

Supplementary Figure 3: A color legend is provided in the graphs b and d, but no colors are presented in the actual bars from the graph. This should be adapted.

Line 142: the authors describe many features are revealed my M3CN. Although the example of IgH is provided, this termination remains vague and should be better explained. Are there links with the known molecular subgroups for example?

Author Response

Please see the response to reviewers comments letter.

Best,

-Jun

Reviewer 2 Report

In this manuscript, Liu and colleagues use publically available RNAseq data from a large cohort of multiple myeloma (MM) patients to generate a map of gene interactions that may play a role in MM disease progression. The paper is well written, clear and logically argued. The authors’ approach of incorporating CNV and transcriptome analyses to differentiate cis-CNV regulated genes with biological regulations is an interesting one which appears to be informative. The integration into this data into the Multiple Myeloma Molecular Causal Network (M3CN) has revealed interesting potential interactions that may have biological relevance. However, some additional analyses could significantly strengthen the manuscript.

To date, there are a large number of published transcriptomic studies which have attempted to identify gene signatures that could inform outcomes and treatment decisions for patients. However, as acknowledged by the authors of the current study, there is little overlap in the identified signatures between these different studies, making their broader application as predictive biomarkers difficult. The authors have found that their M3CN signature, similarly, has few overlaps with previously published prognostic signatures. One shortcoming of this current analysis is that the authors have not validated their approach in one (or more) independent datasets. Without independent validation of their approach, it is difficult to see why their signature has any advantages, or is more widely applicable, that any of the previously published signatures.

The authors infer that their analyses could be useful for determining prognosis associated with different therapies, to potentially enable personalised treatment decisions. As survival and treatment data is available associated with the MMRC dataset, the manuscript could be significantly strengthened by directly analysing the association between the signature identified and outcomes for patients on different therapies.

MINOR ISSUES

Is there an issue with identifying potential issues with annotation by aligning the annotated gender with the “inferred gender” based on expression of X and Y chromosome specific genes? It may be that an individual’s self-reported gender identity is different from their sex based on genetics, but this does not necessarily infer that this is an annotation error (ie. The individual may not be cis)

Author Response

(The authors gave the same response as above.)

Round 2

Reviewer 1 Report

The authors have sufficiently addressed the comments.

Perhaps I missed it somewhere but can the authors provide the password for opening the supplementary files as I am requested to fill in a password?

Minor issue:

As the authors cannot prove that that their network is able to predict treatment responses, there statement about their findings can be used for precision medicine should be rephrased or removed.

Author Response

Please see the attached response to reviewer comments file.

Reviewer 2 Report

The authors have significantly altered their manuscript to address many of my concerns.

Three minor issues remain, that should be corrected before the manuscript is suitable for publication.

While the authors have gone a way towards assessing the prognostic significance of the genes in their prognostic subnetwork in an independent cohort, they still have not definitively demonstrated that their signature reveals differences in response to specific therapies. In the absence of this more comprehensive assessment, the statement that their analysis provides “biological insights for precision medicine” (in the abstract and intro) is not entirely supported by their data, as they have not demonstrated that this signature can be used for clinical decision making with regard to treatment. As such, they should remove this statement from the manuscript. While I appreciate the authors re-wording their analyses to more correctly talk to the sex of the individual, rather than the gender, as inferred by X and y gene expression, I still feel that this is an inappropriate way of defining inaccuracies in the annotation. As I said previously, these discrepancies could arise because the biological sex of the individual is different from their self-reported gender, rather than being due to incorrect annotation – as such, excluding samples on the basis of these discrepancies may be incorrect. I do note that the sex-discrepancies observed were in samples that also had other issues ie. alignment between mRNA and CNV, heavy chain discrepancies. As such, I understand that the authors are not basing their decision to exclude those samples on sex/gender discrepancies alone. However, as this method of deducing inaccuracies may be misleading, I would suggest that the authors either remove it from the analysis, or discuss it after they discuss the other methods you used to identify discrepancies, to show that the sex/gender mismatches support their mismatch call by other means, and were not the primary method used to identify these samples. Many supplementary tables are provided as text files – these would be more readily readable if they were formatted as tables in a word document or similar and provided as pdf files, or provided as an excel spreadsheet (or similar). It is unwieldy for a reader to have to import the files themselves into their software of choice before being able to assess the data.

Author Response

(The authors gave the same response as above.)
